# Technical evaluation and usability of a quantitative G6PD POC test in cord blood: a mixed-methods study in a low-resource setting

Germana Bancone [1,2] Mary Ellen Gilder,[1] Elsie Win,[1] Gornpan Gornsawun,[1] Penporn Penpitchaporn,[1] Phaw Khu Moo,[1] Laypaw Archasuksan,[1] Nan San Wai,[1] Sylverine Win,[1] Ko Ko Aung,[1] Ahmar Hashmi,[3,4] Borimas Hanboonkunupakarn,[5] Francois Nosten [1,2] Verena Ilona Carrara,[1,6] Rose McGready [1,2]

For numbered affiliations see end of article.

**Correspondence to**
Dr Germana Bancone;
germana@tropmedres.ac

## ABSTRACT

**Objectives** New point-of-care (POC) quantitative G6PD testing devices developed to provide safe radical cure for *Plasmodium vivax* malaria may be used to diagnose G6PD deficiency in newborns at risk of severe neonatal hyperbilirubinaemia, improving clinical care, and preventing related morbidity and mortality.

**Methods** We conducted a mixed-methods study analysing technical performance and usability of the 'STANDARD G6PD' Biosensor when used by trained midwives on cord blood samples at two rural clinics on the Thailand–Myanmar border.

**Results** In 307 cord blood samples, the Biosensor had a sensitivity of 1.000 (95% CI: 0.859 to 1.000) and a specificity of 0.993 (95% CI: 0.971 to 0.999) as compared with gold-standard spectrophotometry to diagnose G6PD-deficient newborns using a receiver operating characteristic (ROC) analysis-derived threshold of ≤4.8 IU/gHb. The Biosensor had a sensitivity of 0.727 (95% CI: 0.498 to 0.893) and specificity of 0.933 (95% CI: 0.876 to 0.969) for 30%–70% activity range in girls using ROC analysis-derived range of 4.9–9.9 IU/gHb. These thresholds allowed identification of all G6PD-deficient neonates and 80% of female neonates with intermediate phenotypes.

Need of phototherapy treatment for neonatal hyperbilirubinaemia was higher in neonates with deficient and intermediate phenotypes as diagnosed by either reference spectrophotometry or Biosensor.

Focus group discussions found high levels of learnability, willingness, satisfaction and suitability for the Biosensor in this setting. The staff valued the capacity of the Biosensor to identify newborns with G6PD deficiency early ('We can know that early, we can counsel the parents about the chances of their children getting jaundice') and at the POC, including in more rural settings ('Because we can know the right result of the G6PD deficiency in a short time, especially for the clinic which does not have a lab').

**Conclusions** The Biosensor is a suitable tool in this resource-constrained setting to identify newborns with abnormal G6PD phenotypes at increased risk of neonatal hyperbilirubinaemia.

## STRENGTHS AND LIMITATIONS OF THIS STUDY

⇒ The technical performance of the G6PD quantitative point-of-care diagnostic device was assessed against the current gold-standard spectrophotometric assay.

⇒ Receiver operating characteristic analysis was used to identify the best diagnostic thresholds.

⇒ Usability among clinical personnel from a resource-constrained setting was analysed using a conceptual framework developed for similar settings.

⇒ Fewer than planned focus group discussions were conducted and they occurred in a single clinical site providing a possibly narrower point of view on the usability topics explored.

## INTRODUCTION

Pathologically increased levels of bilirubin during the first week of life, that is, neonatal hyperbilirubinaemia (NH), are common and dangerous for the developing brain. The most severe form of NH, kernicterus, causes neurological sequelae in >80% of neonates (56/100 000 live births globally[1]). Every year, an estimated 24 million newborns are at risk of NH-related adverse outcomes with three-quarters of mortality occurring in sub-Saharan Africa and South Asia.[1 2] These preventable deaths and disabilities disproportionally affect neonates where universal healthcare and treatment options are scarce, if not absent.[3]

Several genetic and clinical factors influence the timing and evolution of NH, including G6PD deficiency, ABO blood group incompatibility, prematurity/low birth weight and sepsis.[4] Early identification of these risk factors can dramatically improve neonatal clinical management during the first days of life.[5]

The enzymatic defect of G6PD deficiency, caused by mutations on the X-linked G6PD gene, is a known risk factor for increased levels of bilirubin after birth and it is associated with susceptibility to drug-induced haemolysis.[6] Risk of severe NH is increased in both deficient and heterozygous newborns with abnormal phenotypes,[7–9] and universal neonatal screening of G6PD deficiency is supported by WHO in populations where more than 3%–5% of boys are affected.[10]

G6PD deficiency is particularly prevalent among neonates from tropical regions,[11] where clinical care is often provided in a non-tertiary hospital or clinic context. Knowledge of G6PD status by medical staff and parents can aid in avoiding potentially haemolytic antibiotics or other agents (such as naphthalene), improved follow-up, and heightened awareness of signs and symptoms of severe NH.

G6PD deficiency is very common among the Karen and Burman populations along the Thailand–Myanmar border (9%–18% in males[12]), where it is associated with an increased risk of developing NH requiring phototherapy both in G6PD-deficient (over fourfold[13]) and in heterozygous females (over twofold[5]) as compared with wild-type genotype neonates. In a recent study, screening of G6PD by qualitative fluorescent spot test (FST) on cord blood failed to identify almost 10% of G6PD-deficient neonates.[14]

Demonstrating usability of new quantitative point-of-care (POC) G6PD diagnostic tests by locally trained clinical staff can inform clinical deployment in this setting and in other rural settings. This study assessed the technical performance and usability of the 'STANDARD G6PD' (SD Biosensor, Korea) test when used by trained midwives in two clinics along the Thailand–Myanmar border.

## MATERIALS AND METHODS
### Study design
A mixed-methods study was conducted to evaluate both the technical performance of the 'STANDARD G6PD' test (henceforth 'Biosensor') and its usability by midwives in a non-tertiary setting. G6PD enzymatic activity and haemoglobin concentration measured by the device were compared with the gold-standard reference spectrophotometric assay and haematology analyser, respectively. Performance of the G6PD FST currently used routinely at the POC was also compared with the reference and new test.

Following local staff training, user proficiency was assessed before study start; usability was explored using focus group discussions (FGDs) at the end of the study.

### Study setting and population
The study was conducted in Shoklo Malaria Research Unit (SMRU) clinics situated along the Thailand–Myanmar border in Tak province (Thailand) where free antenatal care and birthing services are provided for migrant women of predominantly Karen and Burman ethnicity.

SMRU midwives come from the same population as the pregnant women and patients seeking care at SMRU clinics. The majority of midwives have primary or secondary education and receive clinical training on-site. Pregnant women attending SMRU clinics at Wang Pha (WPA) and Maw Ker Thai (MKT) were informed about the study at regular antenatal care visits in the third trimester. Informed consent procedures and eligibility assessments for mothers were completed before labour commenced. Eligibility of neonates was assessed immediately after delivery, and those born at an estimated gestational age (EGA) by ultrasound ≥35 weeks with no severe maternal complications at delivery and no severe neonatal illness were included. In order to allow laboratory analyses to be performed within 30 hours from collection, only neonates born during weekdays were included. For all neonates, indication for starting phototherapy treatment followed the recommendations of the UK National Institute for Health and Care Excellence (NICE) guidelines.[15]

### Blood analyses for technical evaluation of Biosensor
Two millilitres of cord blood were collected into EDTA from the umbilical cord using an established SMRU standard operating procedure (SOP). An aliquot of anticoagulated blood was used by the midwives in the delivery room for the Biosensor following manufacturer's instructions within 1 hour of collection (online supplemental file 1). Tests were repeated if the test result was an error or 'HI' (a result obtained when G6PD activity is very high, outside the instrument analytical range). High-level and low-level Biosensor controls were run weekly or monthly (depending on availability) from April 2020 until May 2021.

An aliquot of anticoagulated blood was analysed by G6PD FST at the clinical laboratory. The remaining blood was stored at 4°C until shipment to the central SMRU laboratory on the same day.

Gold-standard reference testing for G6PD and haemoglobin was performed by spectrophotometric assay and haematology analyser (with complete blood and reticulocyte counts), respectively, at the SMRU central laboratory.

G6PD spectrophotometric assay was performed using Pointe Scientific kits (assay kit # G7583-180, lysis buffer # G7583-LysSB). Kinetic determination of G6PD activity at 340 nm was performed using a SHIMAZU UV-1800 spectrophotometer with temperature-controlled cuvette compartment (30°C). Samples were analysed in double and mean activity was expressed in IU/gHb using the haemoglobin concentration obtained by complete blood count analysis. The final result was calculated using manufacturer's temperature control factor of 1.37. Two controls (normal, intermediate or deficient; Analytic Control Systems, USA) were analysed at every run and results compared with expected ranges provided by manufacturer. Complete blood count was performed using a CeltacF MEK-8222K haematology analyser (Nihon Kohden, Japan). Three-level quality controls were run every day, and device maintenance and calibration were

performed regularly. Reticulocytes were analysed by microscopy after staining with supervital staining Crystal Violet.

Buffy coat recovered from whole blood after centrifugation was stored at −20°C for later DNA extraction using standard columns kit (Favorgen Biotech, Taiwan). Genotyping for G6PD common mutations was performed through established SOPs.[16] Mahidol mutation was analysed in all samples. Other mutations were only analysed in phenotypically deficient or intermediate samples (G6PD<9.31 IU/gHb by reference test) with wild-type or heterozygote Mahidol genotypes. Viangchan, Chinese-4, Kaiping, Canton, Union and Mediterranean were analysed first, and full gene sequence was performed if none of these mutations were found.

### Biosensor training, user proficiency and usability assessment

Midwives of WPA and MKT SMRU clinics were trained for use of Biosensor and were eligible to participate in the usability component of the study following informed consent. Two to four training sessions were provided at each clinic in the local language by an experienced laboratory technician (author LA). The sessions lasted from 1 to 2 hours and included a short introduction about the test, a practical demonstration using imitation blood and supervised use of the Biosensor by each midwife. Midwives were allowed to practise the procedure the week following the training prior to taking a user proficiency test. The proficiency test was administered by author LA in the local language and it consisted of a questionnaire (modified from a questionnaire developed by PATH (https://www.finddx.org/wp-content/uploads/2020/09/PATH_STANDARD-G6PD-User-Competency-Assessment-quiz_08oct19.pdf)) and direct observation of two consecutive tests. Midwives were asked to explain out loud their actions while performing the first test. The proficiency test was analysed by authors GB and GG and midwives who scored <85% were retrained before study start. A visual aid with all critical steps of the procedure was printed and available in the delivery room during the study.

The usability component of the study followed the conceptual framework for acceptance and use of a rapid diagnostic test for malaria proposed by Asiimwe et al[17] that evaluates six components: learnability, willingness, suitability, satisfaction, efficacy and effectiveness. The FGDs specifically focused on four main themes of learnability, willingness, satisfaction and suitability. Due to COVID-19, only two of the planned six total FGDs were conducted. The midwives were grouped by their seniority, with senior and junior midwives together, and midwife assistants in a separate group in order to encourage honest and open conversation. One researcher (KKA) facilitated the FGD, while an experienced assistant took notes; both were fluent in Burmese and Karen languages used in the FGD. Immediately following the FGD, research staff debriefed and noted main themes of the discussion. FGDs were audio-recorded and subsequently translated and transcribed in English. Two researchers (MEG and GB) independently analysed the transcript using thematic analysis based on the preset framework[17] using Taguette (a free and open-access qualitative data analysis software, https://joss.theoj.org/papers/10.21105/joss.03522) and confirmed findings with KKA. Face-to-face meeting and exchange of notes allowed for triangulation between the researchers.

### Blood analysis for assessment of NH

Routine clinical care for newborns included at least one total serum bilirubin test before discharge (around 48 hours of life) using capillary blood measured on-site by the rapid quantitative bilirubinometer BR-501 (Apel Co, Japan).

### Sample size and statistical analyses

The expected prevalence of G6PD deficiency in the population living at the border is 9%–18% in male and 2%–4% in female,[12 16] corresponding to approximately 20%–30% heterozygous girls, 60% of whom have intermediate activity.[18] Assuming that the proportion of girls and boys in the neonate population is 50%, 9% were expected to be G6PD deficient and 7% to be G6PD intermediate. In order to obtain 95% CI of the limits of agreement (LoA) within 0.5 SD of the difference, about 31 neonates with deficiency and 25 with intermediate phenotypes were needed, with a minimum total sample size of 350 samples.

Clinical data were double entered in MACRO and collated with laboratory data; data were analysed using SPSS V.27.

Male median (MM) was calculated in all boys with wild-type genotypes in both the references spectrophotometric assay and the Biosensor. Deficiency was defined as enzymatic activity below 30% of MM by reference spectrophotometry and receiver operating characteristic (ROC)-derived 30% threshold by Biosensor; intermediate phenotypes were defined as enzymatic activity between 30% and 70% of the MM or ROC-derived threshold.

Mean and SD were reported for continuous variables. Categorical variables were compared by $X^2$ test and analysis of variance. Bland-Altman plot was used to inspect correspondence between G6PD activity detected by Biosensor compared with the spectrophotometry assay.[19] Correlation was assessed using Pearson's coefficient of correlation and interclass correlation coefficient (ICC). Area under the curve (AUC) of the ROC curve[20] was calculated at different activity thresholds to analyse clinical performances (ie, sensitivity and specificity) of the Biosensor. Cohen's kappa coefficient was calculated for categories of phenotypes identified by Biosensor and spectrophotometry.

For analysis of haematological features and risk of NH, neonates' gestational ages assessed by ultrasound were categorised as ≤38 and >38 weeks according to epidemiological studies conducted previously in the same population.[21]

Statistical significance was assessed at the 5% level.

**Table 1** Haematological characteristics of cord blood samples according to newborn gestational age

| EGA (weeks) | N* | WBC (10⁹/L) | NEU (10⁹/L) | LYM (10⁹/L) | RBC (10¹²/L) | HGB (g/L) | HCT (%) | MCV (fL) | MCH (pg) | MCHC (g/dL) | RDW (%) | PLT (10³/µL) | Reticulocyte (%) |
|---|---|---|---|---|---|---|---|---|---|---|---|---|---|
| <38 | 19 | 13.1 (3.6) | 9.6 (3.3) | 2.7 (1.7) | 4.3 (0.4) | 144 (17) | 48.0 (5.4) | 110.9 (6.6) | 33.2 (2.7) | 29.9 (1.5) | 16.8 (1.5) | 259.2 (66.2) | 2.8 (1.8) |
| ≥38 | 298 | 14.3 (3.8) | 10.8 (3.6) | 2.8 (1.6) | 4.5 (0.5) | 145 (17) | 49.0 (5.2) | 109.0 (7.9) | 32.3 (3.0) | 29.6 (1.4) | 16.0 (1.2) | 261.4 (47.7) | 2.1 (1.1) |
| P value$_{ANOVA}$ | | 0.17 | 0.16 | 0.88 | 0.14 | 0.68 | 0.43 | 0.30 | 0.21 | 0.41 | 0.01 | 0.85 | 0.02 |

Results are shown as mean (SD).
*Number of samples analysed by haematology analyser was 317 out of 325; 7 samples were analysed by Hemocue and result used to calculate G6PD enzymatic activity.
ANOVA, analysis of variance; EGA, estimated gestational age; HCT, haematocrit; HGB, haemoglobin; LYM, lymphocyte; MCH, mean corpuscular haemoglobin; MCHC, mean corpuscular haemoglobin concentration; MCV, mean corpuscular volume; NEU, Neutrophil; PLT, platelet; RBC, red blood cell; RDW, red cell distribution width; WBC, white blood cell.

### Patient and public involvement

At the outset of the study, the research team engaged the local population through a local ethics and research advisory committee, the Tak Province Community Advisory Board, Thailand. This group is comprised of community leaders who were asked to advise on study design, process and outcomes of interest, and subsequently approved the study.

### RESULTS

A total of 331 cord blood samples were collected between April 2020 and November 2021; 6 were clotted and excluded from all analyses. Of the remaining 325 samples, 257 (79%) were collected in MKT clinic and 68 in WPA clinic, in 166 (51%) female and 159 male neonates. Mean (SD) of EGA of newborns was 39.1 (1.0) weeks.

### General haematological characteristics

As expected for this specimen, haematological characteristics of cord blood (table 1) showed higher white cell count, haemoglobin concentrations, reticulocyte counts and larger cellular volumes compared with adult blood. Reticulocyte counts and red cell distribution width were higher in neonates <38 weeks' gestational age (p=0.02 and p=0.01, respectively), while the other indexes did not differ by gestational age groups.

### G6PD genotypes

A total of 26 hemizygous mutated boys (21 Mahidol, 2 Kaiping, 1 Viangchan, 1 Coimbra, 1 Orissa), 3 homozygous mutated girls (Mahidol), 34 heterozygous girls (32 Mahidol, 1 Canton, 1 Viangchan) and 262 wild type (129 girls and 133 boys) were found. Overall, allelic frequency of all mutated alleles was 13.4%. The distribution of G6PD activity by spectrophotometry and Biosensor associated with different genotypes is shown in figure 1 and online supplemental tables 1 and 2.

### Fluorescent spot test

The poor performance of the FST in cord blood was confirmed here, with the FST failing to identify 23% (7 of 30) of deficient neonates and 100% of the intermediate girls (22 of 22; table 2).

### Technical evaluation of Biosensor

#### MMs by reference spectrophotometric assay and Biosensor

MM G6PD activity by spectrophotometer was 13.3 IU/gHb giving a 30% threshold of 4.0 IU/gHb for diagnosis of deficiency; intermediate activity (30%–70%) in girls ranged between 4.1 and 9.3 IU/gHb. The cord blood-specific 30% spectrophotometric threshold identified all the hemizygous male and homozygous female newborns (figure 1A).

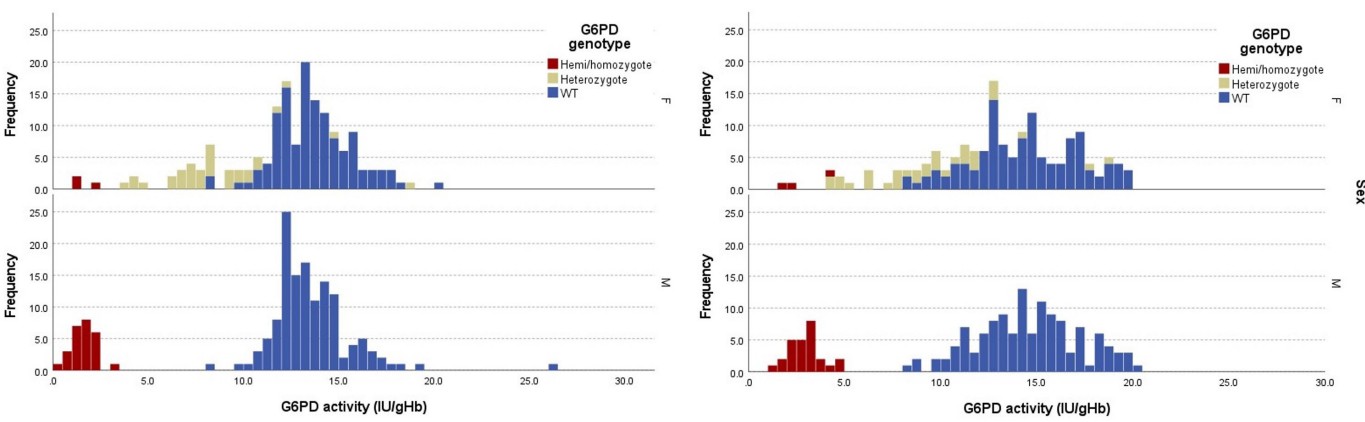

**Figure 1** Distribution of G6PD enzymatic activity from cord blood samples detected by gold-standard spectrophotometry assay (A) and Biosensor (B) according to sex and genotype. WT, wild type.

**Table 2** Diagnostic performance of FST and Biosensor as compared with gold-standard spectrophotometry

| | | Spectrophotometry | | | | |
|---|---|---|---|---|---|---|
| | | **Male** | | **Female** | | |
| | | **Deficient** | **Normal** | **Deficient** | **Intermediate** | **Normal** |
| FST | Deficient | 20 | 0 | 2 | 0 | 0 |
| | Normal | 6* | 133 | 2* | 22 | 137 |
| | Total | **26** | **133** | **4** | **22** | **137** |
| Biosensor | Deficient | 26 | 0 | 4 | 2† | 0 |
| | Intermediate | NA | NA | 0 | 16 | 9‡ |
| | Normal | 0 | 125 | 0 | 4§ | 121 |
| | Total | **26** | **125** | **4** | **22** | **130** |

Phenotypes are based on 30% and 70% thresholds for spectrophotometry. For Biosensor, threshold for deficiency is ≤4.8 IU/gHb and 4.9–9.9 IU/gHb for intermediate, both obtained by ROC analysis.
Total sample for Biosensor was 307; total sample for FST was 322 (3 samples were not analysed by FST at the clinic).
Characteristics of discordant samples are reported in online supplemental table 1.
*Enzymatic activities ranging from 12% to 27% of spectrophotometry MM.
†Two Mahidol heterozygotes with activity by spectrophotometry of 33% and 62% of MM.
‡Two Mahidol heterozygotes and seven wild-type samples with enzymatic activity by spectrophotometry ranging from 71% to 113%.
§Three Mahidol heterozygotes and one wild-type samples with enzymatic activity by spectrophotometry ranging from 54% to 64%.
FST, fluorescent spot test; MM, male median; NA, Not applicable; ROC, receiver operating characteristic.

MM of G6PD activity by Biosensor calculated on 307 samples was 14.4 IU/gHb giving a 30% threshold of 4.3 IU/gHb for diagnosis of deficiency. Intermediate activity (30%–70%) in girls ranged between 4.4 and 10.1 IU/gHb (figure 1B).

In 7% of cases (23 of 325), the Biosensor provided an initial result of 'HI' activity without a numerical value. Of the 19 samples retested, 14 had 'HI' results again and 5 samples had an activity ranging from 17.3 to 20.0 IU/gHb; all samples with initial or confirmed 'HI' results were normal by spectrophotometry and had a wild-type genotype. Overall, 18 samples (5.5% of the total) did not have a final numerical result by Biosensor but would have been considered 'normal', according to the spectrophotometric assay.

### Biosensor performance

Biosensor performance was assessed for 307 of 325 samples that yielded numerical results. The mean (±1.96 SD) difference in enzymatic activity between Biosensor and spectrophotometry was 1.05 IU/gHb (LoA: −3.52 to 5.62 IU/gHb) as represented in the Bland-Altman plot in figure 2A. A very strong correlation between enzymatic activity by Biosensor and reference spectrophotometry was observed (Pearson's r=0.855, p<0.001; ICC=0.905, p<0.001).

The mean (±1.96 SD) difference in haemoglobin between the Biosensor and haematology analyser was 0.70 g/dL (LoA: −2.83 to 4.23 g/dL) (figure 2B). A moderate correlation between haemoglobin levels by Biosensor and

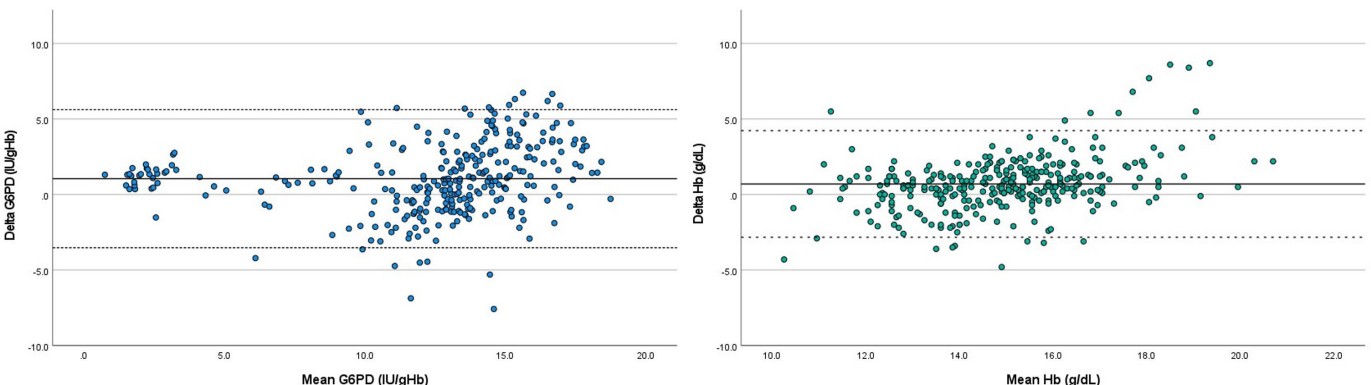

**Figure 2** Bland-Altman plot of G6PD activity (A) and haemoglobin (Hb) levels (B) in cord blood comparing gold-standard spectrophotometry with Biosensor. (A) Delta G6PD=G6PD Biosensor–G6PD spectrophotometry. Full horizontal line indicates mean difference (1.05 IU/gHb); dotted horizontal lines indicate limits of agreement (−3.52 to 5.62 IU/gHb). (B) Delta Hb=Hb Biosensor–Hb spectrophotometry. Full horizontal line indicates mean difference (0.70 g/dL); dotted horizontal lines indicate limits of agreement (−2.83 to 4.23 g/dL).

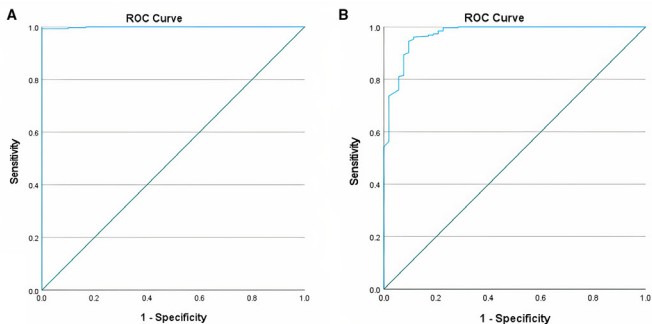

**Figure 3** Receiver operating characteristic (ROC) curve of Biosensor for 30% activity (A) and 70% activity (B) thresholds.

haematology analyser was observed (Pearson's r=0.637, p<0.001; ICC=0.728, p<0.001).

AUC of the ROC analysis (figure 3A) of the 30% threshold was 0.999 (95% CI: 0.997 to 1.000); ROC analysis showed that 30% of Biosensor MM (4.3 IU/gHb) was associated with sensitivity of 0.931 (95% CI: 0.758 to 0.988) and specificity of 0.989 (95% CI: 0.966 to 0.997), while a threshold of 4.8 IU/gHb had a sensitivity of 1.000 (95% CI: 0.859 to 1.000) and a specificity of 0.993 (95% CI: 0.971 to 0.999). This second threshold was therefore used for the subsequent analyses.

AUC of the ROC analysis (figure 3B) for the 70% threshold was 0.972 (95% CI: 0.949 to 0.994) and ROC analysis showed that a threshold of 9.9 IU/gHb had a better sensitivity and specificity as compared with the 70% of Biosensor MM (10.1 IU/gHb). The ROC-derived threshold had a sensitivity of 0.842 (95% CI: 0.716 to 0.921) and specificity of 0.984 (95% CI: 0.957 to 0.995) to identify samples with ≤70% activity and was used for subsequent analyses.

AUC of the ROC analysis for the range 30%–70% activity was 0.935 (95% CI: 0.887 to 0.983); sensitivity and specificity for intermediate phenotypes in girls were 0.727 (95% CI: 0.498 to 0.893) and 0.933 (95% CI: 0.876 to 0.969), respectively, based on ROC-derived thresholds as compared with 0.592 (95% CI: 0.390 to 0.770) and 0.953 (95% CI: 0.897 to 0.980) using Biosensor MM thresholds.

When comparing phenotypes defined according to the 30% and 70% thresholds of spectrophotometry and ROC-derived threshold for Biosensor (table 2), the Biosensor correctly identified all deficient and normal boys and all deficient girls. In girls, the Biosensor incorrectly identified 9% (2 of 22) of intermediate girls (activity by spectrophotometry 33% and 62%) as deficient, and 7% (9 of 130) of phenotypically normal female neonates as intermediate (activity by spectrophotometer ranging from 71% to 113%). It also misdiagnosed 18% (4 of 22) of intermediate samples as normal. Of these four samples, three were Mahidol heterozygotes and one was a wild type, and their enzymatic activity by spectrophotometry ranged from 54% to 64%. Cohen's kappa coefficient was 0.841 (p<0.001). Overall, the majority of samples with discordant results (11 of 15) were identified by the Biosensor as having a 'worse' phenotype. Characteristics of the 15 samples with discordant results are reported in online supplemental table 3.

No difference in results was observed by clinic (ICC=0.899, p<0.001 in MKT and ICC=0.930, p<0.001 in WPA) or user. In MKT clinic where the test was used over 20 months, a trend of larger absolute mean differences in activity (Biosensor–spectrophotometry) was observed in the last 4–8 months of use as compared with the first 12 months (online supplemental figure 1).

### Risk of NH

Risk of NH by phenotype (determined by spectrophotometry) was assessed in term neonates (EGA ≥38 weeks). A significantly larger proportion of G6PD-deficient neonates (29%) underwent phototherapy for treatment of NH as compared with G6PD normal (6%, relative risk (RR) (95% CI)=4.9 (2.3 to 10.5); p<0.001). A larger proportion of female neonates with intermediate phenotypes (90% of whom were heterozygotes) required phototherapy (15%), although in this small cohort, the difference did not reach statistical significance (RR (95% CI)=2.6 (0.8 to 8.1); p=0.13; online supplemental table 4). RRs by quantitative phenotypes were similar to those already established by genotypes in the same population.[5]

### Biosensor training, user proficiency and usability assessment

A total of 22 midwives in two clinics were initially trained and completed the users' proficiency test, including 7 senior, 10 junior and 5 assistant midwives. Median (min–max) observed score from the questionnaire (max 7 points) and observed tests (max 18 points) was 22.1 (18–24.5). The median score did not differ by seniority: assistant 21.4 (18.0–23.5), junior 22.0 (19.3–24.5), senior 22.8 (21.0–24.5); most midwives (72%) had a score >21 points (>85% of maximum score). The most common mistakes in the questionnaire were on how to mix the blood and the buffer (pipetting 10 times vs shaking the buffer tube) and on volume of blood mixture to transfer into the device. On observation, the most common mistakes were failure to check the date on Biosensor screen and failure to check test expiry date (rated as minor mistakes since expired test strips are automatically recognised by the Biosensor and rejected).

Two FGDs were held in December 2021 in MKT clinic, 4 weeks after completion of the sample collection at that site; one FGD included six senior and junior midwives, and one included six assistant midwives. Discussions on satisfaction, learnability, willingness, and suitability and future use are summarised in table 3. Overall satisfaction was high, although staff were concerned with invalid results, and found it challenging to dedicate one member of the team to perform the Biosensor test in the delivery room in the busy postpartum period. In terms of learnability, the midwife assistants reported learning the device more easily, though some were anxious about missing steps. The senior staff were anxious about mistakes and clotted blood, and reported the need to refer to the instructions

**Table 3** Selected quotes by theme from focus group discussions (FGDs)

| Theme | Quotes |
|---|---|
| A. Satisfaction | "It is very good for the children. It is good to know if the child has G6PD deficiency or not from birth. The advantage of the device is that it can detect the children without having to do a heel stick on the baby. On the other hand, there is an increase in work…. But now that we are good at using it, it's fine." (FGD1)<br>"Sometimes if someone is doing the test by using the device it means there are fewer staffs to be with mothers and babies which is not good." (FGD1) |
| B. Learnability | "After the one-time training, we had 1 or 2 times experiences practically. Then we can do it." (FGD2)<br>"I am really scared I will forget the steps." (FGD2)<br>"We have to look at the book very often, if not we forget the process of what to put and how to put it." (FGD1) |
| C. Willingness | "Facilitator: Yes. What do you think about keeping on using this device in the future?<br>Participant: Of course. It is good.<br>Participant: Yes, it is good. But if we can have a specific staff to do it then it will be better." (FGD2)<br>"To make changes, take out the blood and send it to the lab. Then only lab staff have to do that." (FGD1) |
| D. Suitability and future use | "Because we can know that early, we can have counseling with the parents about the chances of their children getting yellow skin. We can take time to counsel." (FGD1)<br>"Because we can know the right result of the G6PD deficiency in a short time. Especially for the clinic which doesn't have a lab then it is difficult to know the G6PD status. But with this device, they will only need to take a little blood from the baby and they can know the result of G6PD." (FGD2) |

as a problem. Contrary to the positive expressions to keep using the device at the clinic, the midwives' willingness to use the device was not high and they requested a dedicated staff to perform the test or the test to be done in the laboratory. In terms of suitability and future use, the midwives found the results clinically useful and a valuable diagnostic tool in both their setting and field clinics. However, they were concerned about neglecting clinical care while doing a laboratory test, the cost of the device and emphasised the need for good training.

## DISCUSSION

This is the first study to assess clinical performance and usability by locally trained health workers of the 'STANDARD G6PD' Biosensor test for identification of G6PD-deficient and intermediate phenotypes in cord blood. Current data, together with previously collected evidence from clinical trials in the same population,[5] clearly indicate that newborn heterozygous girls with G6PD intermediate phenotypes, who are not identified by the FST, are at increased risk of NH and require phototherapy.[7 8] The availability of a validated POC quantitative test such as the Biosensor and its inclusion in diagnostics guidelines for neonatal care at birth will allow identification of this group of neonates and better clinical care in several settings.[22–25] Together with other easy-to-use non-invasive tools for diagnosis of NH (eg, transcutaneous bilirubinometers), this study provides evidence that Biosensor could be used in non-tertiary rural settings for identification of neonates who need referral to higher levels of care. In settings where phototherapy is available, this

study indicates that the Biosensor is a better option than FST to support clinical management of neonates. Technical performance of the Biosensor using ROC-derived threshold was comparable with that observed in adult blood in laboratory and field studies.[26–29]

The phenotypical classification provided by the Biosensor was superior to the currently available qualitative test (FST) both for deficient and for intermediate phenotypes. Among intermediate phenotypes, 80% were identified as either deficient or intermediate, allowing a better identification of neonates at potential jaundice risk as compared with the currently used FST-based diagnosis.[14 30] Poor performance of FST can be explained by the higher G6PD enzymatic activity at birth as compared with adulthood[31 32]; this is probably the result of several haematological factors including younger red cell age, increased number of reticulocytes with higher G6PD activity[33 34] and higher white cell count[28] as observed here. Importantly, because of higher enzymatic activity in cord blood, thresholds established in adult blood cannot be used to identify deficient or intermediate phenotypes by either spectrophotometry or Biosensor at birth and would have missed identification of 10% (3 of 29) deficient neonates (2 of 26 deficient boys and 1 of 4 deficient girls) and 86% (19 of 22) intermediate girls.

Biosensor haemoglobin values had a moderate correlation with those assessed by automatic haematology analyser. Although cord (and neonatal) blood samples have higher haemoglobin levels and increased viscosity, Biosensor's performance in measuring G6PD activity was not worse at higher haemoglobin levels.

While the Biosensor provided a numerical result in 94.5% of cases, in few cases an 'error' message or a 'HI' result was obtained which, according to the protocol, required reanalysis of the sample. Samples that tested 'HI' were confirmed to be normal, both phenotypically by spectrophotometry and by genotype (all wild type). In routine practice, it will not be needed to repeat the test in samples showing 'HI' result should the manufacturers include this information in the instructions for use.

The usability component of the study highlighted important themes to be taken into consideration for future use of the Biosensor at birth. The midwives have been involved in previous research regarding neonatal jaundice and appreciated the importance of early G6PD diagnosis to identify newborns most at risk of NH and to facilitate optimal clinical care and parental counselling. The non-invasive nature of cord blood analysis was considered an advantage. In this setting, the SMRU midwives recommended that the test be performed by dedicated staff or by the available laboratory to assure appropriate clinical care is provided to the newborns and mothers; nevertheless, they estimated that in more rural contexts, it may be appropriate for trained birth attendants to perform the test. Of note, midwives considered their reliance on reading the visual aid while performing the test (which is standard practice in laboratories) a weakness and this aspect might need to be taken into account when training clinic field staff. Usability results obtained here might not be generalisable to every other context but there are data being collected in several rural and community-based settings that corroborate ease of use of this device to guide malaria treatment after appropriate training.[26 35 36]

Although midwives felt uncertain about properly conducting the test at the beginning of the study, the laboratory data showed highly accurate results in the first 12 months of use and very good results in the latter 8 months, supporting suitability of the test among healthcare workers without prior experience in diagnostics. Follow-up studies should explore the causes of this slight decrease in quality over time, which could be attributed to environmental or users' factors as well as device durability over >1 year of use in tropical conditions.

## Limitations

A practical limitation of Biosensor testing on cord blood is the extra step needed to collect the blood with a syringe from the cord. A sampling device that collects a fixed volume of blood directly from the cord would streamline the process.

It is very likely that performance and reference ranges observed here in cord blood could apply to neonatal capillary or venous blood collected within the first 24 hours of life, but this was not evaluated during the study.

The study was conducted in a period critically influenced by the COVID-19 pandemic. Travel restrictions resulted in a delayed study start, reduced enrolment in one clinic (WPA) and protracted enrolment duration of the study overall. Fewer than planned FGDs were conducted—including planned discussions at key time points during the study—and they occurred in a single clinical site providing a possibly narrower point of view on the usability topics explored. Additional staff stressors and human resource limitations due to COVID-19 and the political unrest in Myanmar in 2021 were not assessed but may have influenced the results of both the technical and usability components of the study.

## CONCLUSIONS

The 'STANDARD G6PD' Biosensor is a reliable POC tool to support the perinatal care of newborns at higher risk of NH by demonstrating very high sensitivity in identification of deficient newborns and high sensitivity in identification of female newborns with intermediate activity. Its use by trained personnel in rural clinics and birthing centres with a high prevalence of G6PD deficiency, together with assessment of bilirubin levels before discharge, has the potential to avert disability and death from hyperbilirubinaemia.

Extending use of the Biosensor for newborn testing in countries where it is already deployed for malaria case management in resource-constrained settings[37] would provide a higher return on this investment. Use of Biosensor in populations with prevalent G6PD deficiency outside malaria-endemic regions might increase the benefit–cost ratio of universal screening[38] in all settings.[39]

**Author affiliations**
[1]Shoklo Malaria Research Unit, Mahidol-Oxford Tropical Medicine Research Unit, Faculty of Tropical Medicine, Mahidol University, Mae Sot, Thailand
[2]Centre for Tropical Medicine and Global Health, Nuffield Department of Medicine, University of Oxford, Oxford, UK
[3]Institute for Implementation Science, University of Texas Health Sciences Center (UTHealth), Houston, Texas, USA
[4]Department of Health Promotion and Behavioral Sciences, School of Public Health, University of Texas Health Sciences Center (UTHealth), Houston, Texas, USA
[5]Mahidol-Oxford Tropical Medicine Research Unit (MORU), Faculty of Tropical Medicine, Mahidol University, Bangkok, Thailand
[6]Institute of Global Health, Faculty of Medicine, University of Geneva, Geneva, Switzerland

**Acknowledgements** The authors wish to thank all the mothers for their collaboration and understanding; the study would not have been possible without the hard work and dedication of all SMRU staff involved, especially during such a difficult time of political unrest and COVID-19 pandemic. Acknowledgements are also extended to SD Biosensor for donating the devices and the tests for the study.

**Contributors** Substantial contributions to the conception or design of the work—GB, AH, FN, VIC and RM. Acquisition, analysis or interpretation of data for the work—GB, MEG, EW, GG, PP, PKM, LA, NSW, SW, KKA, AH, BH, FN, VIC and RM. Drafting the work or revising it critically for important intellectual content—GB, MEG, EW, GG, PP, PKM, LA, NSW, SW, KKA, AH, BH, FN, VIC and RM. Final approval of the version to be published—all authors. Agreement to be accountable for all aspects of the work in ensuring that questions related to the accuracy or integrity of any part of the work are appropriately investigated and resolved—all authors. Author acting as guarantor—GB.

**Funding** The study was supported by a grant to GB by the Wellcome Trust Institutional Translational Partnership Awards: Thailand Major Overseas Programme (WT-iTP-2019/004). SMRU is supported by the Wellcome Trust (grant 220211). For the purpose of Open Access, the authors have applied a CC BY public copyright licence to any author accepted manuscript version arising from this submission.

**Disclaimer** The funders had no role in study design, data collection and analysis, decision to publish or preparation of the manuscript.

**Competing interests** None declared.

**Patient and public involvement** Patients and/or the public were involved in the design, or conduct, or reporting, or dissemination plans of this research. Refer to the Methods section for further details.

**Patient consent for publication** Not required.

**Ethics approval** This study involves human participants and was approved by the Oxford Tropical Research Ethics Committee, UK (OxTREC 532-19), the Mahidol University Faculty of Tropical Medicine Ethics Committee, Thailand (TMEC 19-048, MUTM 2019-080-02) and the Tak Province Border Community Ethics Advisory Board (TCAB201904).Written informed consent was obtained from literate mothers and midwives; a thumbprint was obtained in the presence of a literate witness for illiterate mothers.

**Provenance and peer review** Not commissioned; externally peer reviewed.

**Data availability statement** De-identified participant data are available from the Mahidol Oxford Tropical Medicine Data Access Committee upon request from this link: https://www.tropmedres.ac/units/moru-bangkok/bioethics-engagement/datasharing.

**ORCID iDs**
Germana Bancone http://orcid.org/0000-0003-4550-0431
Francois Nosten http://orcid.org/0000-0002-7951-0745
Rose McGready http://orcid.org/0000-0003-1621-3257

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
