## [Reviewer comments · BMJ Open]

ARTICLE DETAILS

TITLE (PROVISIONAL)	Technical evaluation and usability of a quantitative G6PD POC test in cord blood: a mixed methods study in a low-resource setting
AUTHORS	Bancone, Germana; Gilder, Mary Ellen; Win, Elsie; Gornsawun, Gornpan; Penpitchaporn, Penporn; Moo, Phaw Khu; Archasuksan, Laypaw; Wai, Nan San; Win, Sylverine; Aung, Ko Ko; Hashmi, Ahmar; Hanboonkunupakarn, Borimas; Nosten, Francois; Carrara, Verena Ilona; McGready, Rose

VERSION 1 – REVIEW

REVIEWER	David Roper
REVIEW RETURNED	07-Sep-2022

GENERAL COMMENTS	The paper is well written and highlights the valuable utility of this point of care (POC) device for the accurate identification of G6PD deficiency and shows it is a marked improvement over the traditional Fluorescent Spot Test (FST) to support clinical management of neonates. It is gratifying that suitability of the POC test among health care workers without prior experience in diagnostics was demonstrated. As stated, the study was conducted in a period critically influenced by the COVID-19 pandemic and this has limited the user-feedback from the “focus group discussions” in terms of learnability, suitability etc for the Biosensor which is a pity. Presumably the same accounts for the slightly reduced sample number, as a minimum total sample size of 350 samples was initially required. The trend of larger absolute mean differences in activity observed in the last 4-8 months of use as compared to the first 12 months is a concern, and hopefully the cause(s) will come to light during the follow up studies.
--

REVIEWER	Tina Slusher University of Minnesota Academic Health Center, Pediatrics
REVIEW RETURNED	26-Sep-2022

GENERAL COMMENTS	I think this is an important well written study and addresses everything well except the cost. One of participants mentioned cost as a limitation and I think it would be extremely important to give the reader and estimate of how affordable this test would be in a low and middle income country. If is cost prohibitive to be used widely that makes this study much less applicable to the population who needs it most.
--

VERSION 1 – AUTHOR RESPONSE

Reviewer: 1

David Roper

Comments to the Author:

The paper is well written and highlights the valuable utility of this point of care (POC) device for the accurate identification of G6PD deficiency and shows it is a marked improvement over the traditional Fluorescent Spot Test (FST) to support clinical management of neonates. It is gratifying that suitability of the POC test among health care workers without prior experience in diagnostics was demonstrated.

As stated, the study was conducted in a period critically influenced by the COVID-19 pandemic and this has limited the user-feedback from the “focus group discussions” in terms of learnability, suitability etc for the Biosensor which is a pity. Presumably the same accounts for the slightly reduced sample number, as a minimum total sample size of 350 samples was initially required.

The trend of larger absolute mean differences in activity observed in the last 4-8 months of use as compared to the first 12 months is a concern, and hopefully the cause(s) will come to light during the follow up studies.

R#1: We would like to thank the reviewer for his comments.

Reviewer: 2

Dr. Tina Slusher, University of Minnesota Academic Health Center

Comments to the Author:

I think this is an important well written study and addresses everything well except the cost. One of participants mentioned cost as a limitation and I think it would be extremely important to give the reader an estimate of how affordable this test would be in a low and middle income country. If its cost prohibitive to be used widely that makes this study much less applicable to the population who needs it most.

R#2: We appreciate the reviewer's comment and the cost of testing is certainly an important issue in LMIC. Nevertheless, it is beyond the scope of the current work so we would like to keep the manuscript as it is.

We are happy however to provide more information here. The current price of the Biosensor analyser in Thailand is 400USD, and each test costs 4 USD but there are some additional considerations about the market in which the Biosensor will play in the near future. First, the test was developed for malaria and it is currently part of Global Found supported tests; we expect that health systems in several LMIC that are already using the Biosensor for malaria will naturally extend its use to newborn testing. Additionally, because the gold standard spectrophotometric assay is not widely available, it is far more expensive (ca 20 USD per test) and requires initial large investments to acquire expensive laboratory equipment (ca 15,000USD), it is expected that the Biosensor will fill a diagnostic gap even in secondary or tertiary hospitals/clinics of middle income countries. Both these scenarios are very likely to drive the Biosensor's price down in a close future.

We think that reporting the mere current price of the Biosensor would not give a fair indication of its applicability in LMIC. We believe that a comprehensive analysis of cost- effectiveness should be provided instead, that includes the cost of the test (with the aforementioned projections for the future) but also the averted costs of decrease morbidities that would be associated with routine use of the test. We are planning to conduct such analysis in collaboration with health economists.

Please note that declarative titles are not part of the journal format. As such, please revise the title of your manuscript to include the research question, study design and setting. This is the preferred format of the journal. See published articles for examples.

The Title has been revised to: "Technical evaluation and usability of a quantitative G6PD POC test in cord blood: a mixed methods study in a low-resource setting".